# Targeting TRPV1 for Cancer Pain Relief: Can It Work?

**DOI:** 10.3390/cancers16030648

**Published:** 2024-02-02

**Authors:** Arpad Szallasi

**Affiliations:** Department of Pathology and Experimental Cancer Research, Semmelweis University, 1085 Budapest, Hungary; szallasi.arpad@semmelweis.hu

**Keywords:** capsaicin, resiniferatoxin, cancer pain, TRPV1 receptor

## Abstract

**Simple Summary:**

Medical control of cancer pain is often unsatisfactory. Narcotic drugs (opioids) are effective pain killers, but they have important negative effects on the central nervous system and the are also highly addictive. A logical strategy to avoid central narcotic adverse effects is to target the peripheral nociceptors where cancer pain is generated. Sensory afferents that express the capsaicin receptor TRPV1 play a central role in cancer pain. In animal experiments, pharmacological blockade or chemical ablation of these nerves provide lasting cancer pain relief. High-dose capsaicin patches are already in clinical use in patients with chemotherapy-induced neuropathic pain. Site-specific resiniferatoxin (an ultrapotent capsaicin analog) injections are currectly undergoing clinical trials in patients with chronic intractable cancer pain caused by metastatic bone disease. This review explores the analgesic potential of small molecule TRPV1 antagonists and the sensory afferent desensitization in cancer patients.

**Abstract:**

Chronic intractable pain affects a large proportion of cancer patients, especially those with metastatic bone disease. Blocking sensory afferents for cancer pain relief represents an attractive alternative to opioids and other drugs acting in the CNS in that sensory nerve blockers are not addictive and do not affect the mental state of the patient. A distinct subpopulation of sensory afferents expresses the capsaicin receptor TRPV1. Intrathecal resiniferatoxin, an ultrapotent capsaicin analog, ablates TRPV1-expressing nerve endings exposed to the cerebrospinal fluid, resulting in permanent analgesia in women with cervical cancer metastasis to the pelvic bone. High-dose capsaicin patches are effective pain killers in patients with chemotherapy-induced peripheral neuropathic pain. However, large gaps remain in our knowledge since the mechanisms by which cancer activates TRPV1 are essentially unknown. Most important, it is not clear whether or not sensory denervation mediated by TRPV1 agonists affects cancer progression. In a murine model of breast cancer, capsaicin desensitization was reported to accelerate progression. By contrast, desensitization mediated by resiniferatoxin was found to block melanoma growth. These observations imply that TRPV1 blockade for pain relief may be indicated for some cancers and contraindicated for others. In this review, we explore the current state of this field and compare the analgesic potential of TRPV1 antagonism and sensory afferent desensitization in cancer patients.

## 1. Introduction

Cancer pain is a general term for a broad range of pain conditions with different etiologies and molecular mechanisms [1,2,3]. It is a serious problem, affecting an estimated one third of cancer patients [4]. In the terminally ill population, the proportion of patients with chronic, intractable cancer pain may exceed 80% [5]. 

Cancer pain is complex and poorly understood [1,2,3,6], hindering drug development. The guidelines for the management of cancer pain were developed by the World Health Organization almost four decades ago [7]. Since cancer pain is not homogenous, satisfactory pain management should be tailored based on an individual assessment of the pain mechanisms. Yet, available treatment options (Table 1) remain symptomatic [8], and for prolonged treatment, patients may become refractory. At present, opioids constitute the mainstay of treatment with worrisome side-effects, like a sedated mental state and constipation, which negatively affect the quality of life of the patients [9]. Even worse, overdosing with opioids can be lethal. Consequently, cancer pain is often undertreated [10]. Clearly, there is a dire need for new treatment modalities with more tolerable side effects. 

Recently, chronic pain has been divided into primary (unknown underlying disease) and secondary pain. Chronic cancer pain is a form of secondary pain; that is, pain linked to underlying disease [11]. The classification of cancer pain is different for solid tumors and hematological malignancies (Table 2 and Table 3) [12,13,14]. In a much simplified manner, pain caused by solid tumors can be divided into three major mechanisms. One, tumor cells can secrete substances that, in turn, activate the tumor-infiltrating sensory nerve ending [15]. Two, cancers can press, infiltrate, or destroy tissues, including sensory nerves [16,17]. And three, metastatic tumors (especially bone metastasis) can create their acidic microenvironment, rich in sensory innervation [18]. Of course, the line between these groups is often blurred. For example, metastatic bone tumors can cause pathological fractures; in this case, generalized bone pain (due to bone marrow infiltration) and bone fracture pain may combine with pain due to activation by the protons of sensory afferents.

Hematological malignancies can cause nociceptive, neuropathic, and mixed pain (Table 3) [14]. The nociceptive group can be further subdivided into superficial somatic, deep somatic, and visceral pain. Neuropathic pain can be central or peripheral. Again, the pain is often mixed. For example, plasma cell myeloma may induce bone pain via osteolysis and fracture, with peripheral neuropathic pain via paraprotein/amyloid production [19]. 

Of note, chemotherapy for cancers can cause neuropathic pain. In fact, the incidence of chemotherapy-induced neuropathic pain (CINP) can be as high as 80% [20]. Sadly, CIPN persists in a large subset of patients even after the discontinuation of the chemotherapy. 

Sensory nerves are attractive candidates to ameliorate cancer pain since they are involved in all major mechanisms (nociceptive, inflammatory, and neuropathic) of cancer pain generation [15,16,17,18]. A major subdivision of primary sensory neurons is characterized by its unique sensitivity to capsaicin [21,22]. These nerves express the capsaicin receptor Transient Receptor Potential Vanilloid 1 (TRPV1) [23], but also carry other TRP channels, like TRPA1 [24,25], as well as non-TRP channels, for example, acid-sensing ion channels (ASICs) [26] implicated in pain perception (Figure 1). This redundancy in pain targets begs the question of whether or not blocking a single receptor, like TRPV1, can provide meaningful pain relief or the whole neuron must be silenced (Figure 2). There are arguments pro and contra for both approaches. In this review, we provide an overview of the current state of this exciting and rapidly changing field, from basic research to clinical trials.

Please note that other TRP channels relevant to cancer pain (e.g., TRPA1 and TRPM8) will be discussed elsewhere in this thematic issue.

## 2. TRPV1 in Cancer Pain: Molecular Mechanisms

TRPV1 is best known as the receptor for capsaicin, the pungent substance in hot pepper [23]. However, TRPV1 is also directly activated by noxious heat [23], a discovery (molecular mechanism of temperature sensation) that earned a shared (with Ardem Patapoutian for the discovery of touch receptors) Nobel prize in Physiology and Medicine for David Julius in 2021. In accord, TRPV1-null mice [28,29], as well as men with non-functioning TRPV1 [30], exhibit deficits in noxious heat sensation. Furthermore, TRPV1 is activated by changes in pH (in particular, by protons) [31,32] and represents a downstream target for various pain-generating substances, as exemplified by bradykinin [33,34].

The cross-talk between cancer cells and sensory afferents in the tumor microenvironment is subject to intensive research [35,36,37,38]. Tumors can actively recruit nerves, and extensive tumor innervation was suggested to herald aggressive disease [39,40,41]. Conversely, the ablation of tumor-infiltrating afferents may ameliorate cancer progression [39,40,41]. Interestingly, cancers display higher electrical activity than normal tissues, and tumors implanted into transgenic mice lacking TRPV1-positive nociceptors neurons show reduced electrical activity [42].

There is preliminary evidence that cancer cells can synthesize substances capable of activating TRPV1. For example, sarcoma cells were shown to produce a lipophilic TRPV1-targeting molecule that is yet to be identified [43]. Tumor cells can also generate formaldehyde, which, in concert with the acidic microenvironment, can synergistically activate TRPV1 [44]. Oral squamous cell carcinoma cells secrete nerve growth factor (NGF) [45], a known regulator of TRPV1 expression [46,47]. NGF can increase TRPV1 protein synthesis, whereas the proto-oncogen Src kinase may promote TRPV1 trafficking into the cell membrane [48,49]. Macrophages in the tumor microenvironment can also activate sensory nerve endings via the interleukin-23 (IL23)/IL17A/TRPV1 axis [50].

The phosphorylation status of the TRPV1 channel protein is, in part, regulated by calcineurin [51], a Ca^2+^ and calmodulin-dependent serine/threonine protein phosphatase. Decreased calcineurin activity is thought to facilitate the transition of acute pain to chronic pain [52,53]. Indeed, tacrolimus (a calcineurin inhibitor) was found to cause severe pain in some transplant patients [53], though it may be mediated by TRPA1 [54], rather than TRPV1 channels. Conversely, restoring calcineurin activity provides pain relief.

In a rodent orthotopic model of breast cancer, dense sensory afferent innervation of the tumor was observed [55]. Co-cultured with sensory neurons, breast cancer cells stimulated neurite outgrowth [56]. This breast cancer-induced aberrant sensory branching may be a major player in breast cancer pain. Of note, melanoma cells can also interact with nociceptive neurons to facilitate neurite growth [57]. The mechanisms by which tumor cells facilitate neurite growth are unknown. One possible candidate is cyclin-dependent like kinase-5 (CDKL5), an enzyme highly expressed in nociceptive neurons [58]. In fact, CDKL5-null mice exhibit defective epidermal innervation and impaired nociception [58]. One may argue that tumor cells promote neurite growth by stimulating CDKL5.

Most patients with metastatic bone cancer experience severe pain. The molecular mechanisms by which bone metastasis generates pain are only beginning to be understood (Figure 1) [59,60,61,62]. Cancer cells create their own acidic microenvironment [62,63,64]. Protons are known activators of TRPV1 [32,65]. In fact, bafilomycin A1, a selective blocker of proton secretion [66], alleviates bone cancer pain [67]. Cancers need a blood supply to grow; therefore, they promote vascular neogenesis (Figure 3) [68,69,70]. Furthermore, osteoblasts can produce insulin-like growth factor-1 (IGF-1) [71]. In a rat model of bone cancer pain, IGF1 was found to up-regulate TRPV1 expression in sensory afferents [72].

TRPV1-expressing nerve endings release calcitonin gene-related peptide (CGRP) that, in turn, can stimulate cancer growth [73,74,75]. In a murine cancer model (Lewis carcinoma inoculated into the paw), tumor growth was attenuated in both TRPV1-null and αCGRP-null mice compared to in wild-type littermates [76]. Thus, a vicious circle is generated in which cancer cells create their own aberrant sensory innervation, and these nerves promote cancer growth.

Most recently, cancer-derived small extracellular vesicles have been implicated in the cross-talk between head-and-neck carcinoma and TRPV1-expressing afferents [76]. The injection of purified cancer-derived vesicles into naïve mice induces hypersensitivity that was absent in TRPV1-null animals [77]. Cancer-derived vesicles also evoke Ca^2+^ uptake in nociceptors, presumably by opening the TRPV1 channel [77].

There is increasing evidence that visceral pain differs from somatic pain. CINP is also fundamentally different from other forms of cancer pain [78,79,80,81]. For example, paclitaxel affects lipid raft formation in sensory neurons [82]. In addition, paclitaxel sensitizes TRPV1 through phosphorylation and increases the number of TRPV1 channels in the lipid rafts [82]. This enhances the chance of an interaction between TRPV1 and toll-like receptor-4 (TLR-4) [82]. Cisplatin [83] and oxaliplatin also stimulate TRPV1 expression [84] (along with TRPA1 [84] and TRPM8 [85]) in sensory afferents.

## 3. Can Selective TRPV1 Antagonism Ameliorate Cancer Pain?

There is increasing evidence that TRPV1 plays a major role in cancer pain [86]. In preclinical studies, selective TRPV1 inactivation through genetic manipulation [87,88] or pharmacological blockade [89] was shown to ameliorate cancer pain. Ehrlich tumor cells injected into the paw of mice cause nociception [90]. The intrathecal administration of AMG9810, a potent small-molecule TRPV1 antagonist, blocks both mechanical and thermal hyperalgesia [90]. The genetic inactivation of *Trpv1* (TRPV1-null mice) also prevents thermal hyperalgesia but has no effect on tumor growth [90]. In a rat model of bone cancer pain (SCC158 carcinoma cells injected into the hind paw), the first generation TRPV1 antagonist capsazepine blocked both mechanical and thermal hyperalgesia [91]. In a follow-up experiment, TRPV1 knock-down via siRNA also ameliorated both mechanical and thermal hyperalgesia [87]. Mice injected with Lewis lung cancer cells experience progressive bone cancer pain, which is markedly reduced in TRPV1-null animals [92].

Taken together, these experiments imply the clinical value of selective TRPV1 blockade in patients with cancer pain.

## 4. Topical Capsaicin Patch for Chemotherapy-Induced Peripheral Neuropathy (CIPN)

Capsaicin is unique among natural compounds in that the initial burning sensation that it evokes is followed by a lasting refractory state (traditionally termed “desensitization”) in which the previously excited sensory neurons are refractory to not only capsaicin but various unrelated chemical and physical stimuli [21,22,93,94,95]. The molecular mechanisms of capsaicin desensitization are only beginning to be understood [22,95,96,97]. Reversible desensitization (sometimes referred to as “functionalization”) should be distinguished from the irreversible loss (chemical ablation) of TRPV1-expressing sensory afferents [95,96].

Strictly speaking, capsaicin desensitization is mediated by the TRPV1 protein. However, at high concentrations, the selectivity of capsaicin for TRPV1 is lost, and capsaicin may start interacting with other targets, including voltage-gated Na^+^ channels [98,99]. In fact, voltage-gated Na^+^ channels are expressed in capsaicin-sensitive dorsal root ganglion (DRG) neurons [100], and capsaicin can be used to deliver the impermeant (permanently charged) sodium channel blocker QX-314 to achieve long-lasting nociceptive blockade [101,102]. However, at present, it is not clear what (if any) role the off-target capsaicin blockade of voltage-gated Na^+^ channels may play in capsaicin-induced analgesia. Regardless of its molecular underpinnings, the analgesic potential of capsaicin desensitization has been confirmed in clinical studies [103,104,105].

High-dose capsaicin patches are well suited for patients with localized pain, such as peripheral neuropathy affecting the feet (Figure 4). In 16 patients with chronic CIPN (mean duration of 2.5 years), a high-dose capsaicin patch (8% capsaicin, Qutenza) applied for 30 min to the affected foot provided the significant reduction of both spontaneous (mean Numeric Pain Rating Scale, −1.27) and evoked pain [106]. The Short-Form McGill questionnaire revealed a reduction if neuropathic pain and the Patient Global Impression of Change showed a significant improvement [106]. Importantly, the treatment restored epidermal nerve fibers in skin biopsies, potentially a “disease-modifying” action [106]. No systemic side-effects were reported.

Post-mastectomy pain affects 25 to 60% of breast cancer patients. It is defined as pain that persists for at least 3 months after surgery. In a pilot study, 12 of the 14 study participants completed a 4-week trial with topical 0.025% capsaicin: eight patients reported satisfactory pain relief [107]. In a follow-up randomized, placebo-controlled trial with 0.075% capsaicin, 5 of the 13 patients were categorized as good responders based on the visual analogue scale for steady pain [108]. An open-label clinical trial with 0.025% capsaicin in Italy reported similar results: out of the 19 study participants, two reported complete pain relief, and an additional 11 patients described a significant reduction in pain [109]. The treatment was well-tolerated with no drop-out due to side-effects. With the 0.075% topical capsaicin cream, clinically meaningful pain relief (53% compared to 17% in the placebo group) was reported in a cohort of 99 cancer patients with post-surgical pain [110]. A recent retrospective analysis of the post-mastectomy pain studies with low-concentration capsaicin creams has identified several problems [111]; therefore, this treatment modality is no longer be recommended.

A high-dose (8%) capsaicin patch is an effective intervention to ameliorate post-surgical pain in general [112] and also in cancer patients [113]. Two case reports described the relief of post-mastectomy pain with Qutenza [114,115]. A multicentric, open, randomized clinical trial is ongoing to compare an 8% capsaicin patch (Qutenza) to per os pregabalin (recommended as an adjuvant analgesic for neuropathic cancer pain [116]) in the early treatment of neuropathic pain after breast surgery [117].

In a French study involving 279 breast cancer patients with peripheral neuropathy caused by surgery, chemotherapy, or radiation therapy, repeated Qutenza applications (on average, 4) resulted in a significant analgesic effect in 82% of the study participants, including those with post-mastectomy pain [118].

In a monocentric observational retrospective real-world data study, in which independent pain physicians completed a Clinician Global Impression of Change survey, significant or complete pain relief was noted in 44% of the 57 participating patients after a total of 184 capsaicin applications [119]. Pain relief was observed after at least three capsaicin applications, and the efficacy increased with repeated treatments. This study also pointed out that capsaicin is not effective in patients whose CIPN was platinum-induced [119]. The most significant side-effect of the capsaicin patch was an initial burning sensation that could be minimized by topical analgesics or cooling of the treated area.

Based on this study, ESMO (European Society for Medical Oncology) has recommended high-dose capsaicin patches as 2nd line therapy for CIPN [120].

The on-going TEC-ORL phase-2 clinical trial (NCT04704453) wishes to compare the analgesic potential of the high-dose (8%) capsaicin patch, Qutenza, and amitryptiline (Laroxyl, the most common analgesic adjuvant used for cancer patients with neuropathic pain [121]) in 130 patients with head-and-neck squamous cell carcinoma [122]. The primary outcome is pain reduction by 2 points during a 9-month trial.

As mentioned above, high-dose capsaicin patches are appropriate for patients with localized, but not generalized, pain. Cancer pain usually affects several dermatomes. Of note, oral cancer patients often experience severe pain at the site of cancer with new sensitivity to spicy food [123]. These patients are very sensitive to local capsaicin challenge [124], indicating increased TRPV1 expression and/or sensitization. It would be interesting to see if desensitization to topical capsaicin could provide pain relief in this patient population.

## 5. Resiniferatoxin for Permanent Cancer Pain Relief: Preclinical Studies

Resiniferatoxin (RTX), isolated from the latex of *Euphorbia resinifera* Berg [125], is an ultrapotent analog of capsaicin with some important differences in pharmacological actions [94]. For example, RTX does not provoke the pulmonary chemoreflex [126], a dose-limiting side-effect of capsaicin administration. Consequently, in the rat full, desensitization of the neurogenic inflammatory response can be achieved by means of a single s.c. RTX administration [94]. The same response can only be replicated with repeated capsaicin administrations over several days.

In the human urinary bladder, RTX evokes a long-lasting (several weeks) but fully reversible desensitization [127,128,129]. This contrasts the irreversible “silencing” action of intrathecal RTX on TRPV1-expressing axons exposed to the cerebrospinal fluid [130,131,132]. This action earned the name “molecular scalpel” for RTX [133].

In the rat, intrathecal RTX (10 to 200 ng administered via lumbar puncture) ablated most TRPV1-positive nerve endings in the dorsal horn of the spinal cord [134,135] and, at the same time, increased the withdrawal latency to radiant noxious heat [134]. In a mouse model of bone cancer pain, intrathecal RTX achieved lasting pain relief [136].

Cancer pain in companion dogs is an important issue in veterinary medicine [137]. Large dogs are prone to develop osteosarcoma in their limbs. These animals experience severe pain and keep the affected, painful limb in an elevated, guarding position. Inspired by the rodent experiments, intrathecal RTX (1.2 μg/kg) was administered to twenty companion dogs with intractable cancer pain under general anesthesia [138]. One hour later the animals were awakened and their vital signs were tested. A transient increase in blood pressure (79 to 131 mmHg) and the heart rate (123 to 161 beats per minute) was noted, which peaked 1–2 h after RTX administration and disappeared by 4 h. The animals also showed a reduced rectal temperature. None of this was really unexpected. Hypothermia is a well-known effect of capsaicin and RTX administration [139]. Furthermore, TRPV1 is expressed in resistance arteries where their activation leads to vasoconstriction, elevating the blood pressure [140]. Importantly, the day after intrathecal RTX administration, the dogs became ambulatory and their owners reported increased comforts levels [138]. RTX did not slow down the progression of the cancer, but the analgesic action persisted until the animals perished [138].

A prospective, randomized, and blinded trial comparing intrathecal RTX (1.2 μg/kg injected into the cisterna magna) with the standard-of-care was carried out on 72 dogs with bone cancer pain [141]. The animal was removed from the trial (“unblinded”) when the owner reported too much pain. More animals in the standard-of-care group (78%) were “unblinded” sooner than those in the RTX group (50%), indicating significant pain relief mediated by RTX [141].

## 6. Resiniferatoxin for Permanent Cancer Pain Relief: Clinical Trials

The favorable experience with intrathecal RTX in companion dogs with bone cancer pain has incentivized the transition to human clinical trials [133,142,143,144]. The first clinical trial with intrathecal RTX (NCT 00804154), by and large, followed the design of the veterinary trial. Patients with intractable pain due to metastatic bone disease (for example, women with cervical carcinoma metastasis to the pelvic bone) were recruited, and RTX was administered intrathecally under general anesthesia (Figure 5) [145]. This open-label, single-site, phase-1 clinical trial (NCT 00804154) involved nine patients. RTX was injected manually at a starting dose of 3 μg in a volume of 1 mL. The first patient reported pain relief at the starting dose, whereas the three other study participants needed a second dose of 13 μg to achieve pain relief. An additional five patients were given a higher RTX dose of 26 μg. Whereas the starting dose was well tolerated, the higher RTX dose resulted in impaired noxious heat sensation as an on-target adverse effect [146]. For example, a few patients suffered scalding injuries by the imbibition of hot coffee. Most of these episodes could be prevented by warning the patients of the danger of hot food and fluids. At present, this study is recruiting patients to test the analgesic potential of an even higher RTX dose (44 μg).

Neuroaxial analgesia includes intrathecal and epidural drug delivery. Since intrathecal (subarachnoid) analgesia is the administration of analgesics directly into the cerebrospinal fluid, drugs given via this route are both much faster and more potent in action. Therefore, it is not unexpected that most opioid side effects are more common and severe if the opioid is administered intrathecally [147]. Similar considerations may apply to neuroaxial RTX treatment [148].

Intrathecal RTX is an effective but problematic means to achieve analgesia. It must be given under general anesthesia because of the transient pain that it evokes, and it puts significant burden on the heart by elevating blood pressure and accelerating the heart rate [114]. Based on the experience with opioids, epidural RTX is expected to lack these complications.

The first clinical trial with epidural RTX (0.4 μg to 25 μg given under mild sedation) enrolled 17 patients (Table 4) [149]. RTX was injected either directly into the epidural space or administered via a catheter placed under fluoroscopic guidance. Three patients reported significant (30%, 50%, and 70%, respectively) reductions in pain scores that lasted until the very end of the study (12 weeks) [149]. Four study participants withdrew from the study or were lost to follow-up, whereas the remaining ten patients died due to the progression of their metastatic disease. At present, Sorrento Therapeutics (San Diego, CA, USA) is recruiting patients for a multicenter, randomized, Phase 2 study to assess the efficacy and safety of a single epidural administration of RTX (15, 20, or 25 μg in 2 mL) versus the placebo for the treatment of intractable pain associated with cancer [150]. This trial was supposed to start in September 2023 with 120 patients.

## 7. Conclusions and Future Research Directions

The complex and poorly understood nature of cancer pain represents a large barrier to drug development [1,2,3,6,151,152]. Available treatment options (Table 1) are symptomatic and cause significant side-effects [8,9]. Sensory afferents represent an attractive alternative to analgesic drugs targeting the CNS, in that drugs that block these nerves are not addictive and do not affect cognition.

A distinct subpopulation of sensory afferents expresses the capsaicin receptor TRPV1 [21,22,153]. There is good evidence both in preclinical [136,138,141] and clinical studies [145,146,149] that TRPV1-positive afferents play an important role in cancer pain. Cancer pain is reduced in TRPV1-null mice compared to in wild-types [76,92]. Moreover, the ablation of TRPV1-positive nerve endings mediated by intrathecal [138,145] or epidural [149] RTX results in lasting pain relief (Figure 5). Intrathecal RTX, however, has to be administered under general anesthesia because of the initial transient pain reaction that it evokes [138,148]. Moreover, intrathecal RTX is associated with significant on-target side-effects, including a spike in blood pressure, increased heart rate, urinary retention, and impaired noxious heat sensation [138]. The optimal dose at which intrathecal RTX provides meaningful analgesia with easily manageable side-effects is yet to be determined.

Epidural RTX can be administered under mild sedation. Furthermore, based on the clinical experience with opioids [147], epidural RTX should be devoid of the adverse effects that complicate the use of intrathecal RTX. This theory is currently being tested in an ongoing clinical trial [150].

In preclinical models of cancer pain, pharmacological blockade [91] or the knock-down of TRPV1 [86,87] also showed analgesic potential. The biggest challenge in analgesic drug development is to determine if a drug that showed promise in animal experiments will also work in human patients. Generally speaking, rodent models are good for acute pain but not so good for chronic pain, like cancer pain [154]. The limited success of translation from preclinical studies to the clinic may reflect our rudimentary understanding of the molecular mechanisms that drive chronic cancer pain [17]. Cancer pain is broadly described as nociceptive, inflammatory, and neuropathic [17], and the animal models were designed accordingly. However, self-perceived pain in cancer patients shows individual differences [155], depending on both the ethnic background [156] and gender [157]. (Parenthetically, RNA sequencing of the trigeminal sensory neurons revealed distinct transcriptomic profiles between male and female mice under tongue-tumor bearing conditions [158].) Furthermore, cancer pain is often associated with anxiety and depression [159], and patients may even experience referred pain at sites not affected by the disease [13]. Therefore, the observation that TRPV1 antagonists ameliorate cancer pain in rodents should be considered with reservations. It has to be kept in mind that the mechanisms by which cancer activates TRPV1 are unknown, therefore it is not clear either whether cancer models mimic human bone cancer pain. As a word of caution, small-molecule TRPV1 antagonists were effective analgesic agents in animal models of inflammatory and neuropathic pain, yet, they failed in clinical trials to relieve migraine or osteoarthritis pain [160,161].

A special situation of cancer pain is oropharyngeal carcinoma, which is amenable to topical treatment. Many patients with tongue or oral carcinoma exhibit increased sensitivity to capsaicin challenge [124]. In the mouse, radiotherapy (20 Gy) evokes mechanical and thermal allodynia via TRPA1 and TRPV1 activation [162]. Furthermore, in animal experiments, radiotherapy-associated oral pain was found to facilitate tumor growth [163]. Therefore, one may argue that the relief of oral pain mediated by capsaicin may also prevent early cancer relapse.

High-dose (8%) capsaicin patches represent a 2nd line of treatment for chemotherapy-induced neuropathic pain [120]. They also show promise for cancer patients with post-operative pain, including post-mastectomy pain [114] and pain that develops after melanoma surgery [113]. The injectable RTX [164] and capsaicin [165] preparations are yet to be tested in this patient population.

One third of healthy individuals reports no pain response to topical capsaicin (1%) challenge [166]. There is increasing evidence that a *TRPV1* gene polymorphism is responsible for individual differences in human sensitivity to capsaicin [167,168]. Therefore, the one-size-fits all approach may not be applicable to capsaicin desensitization: depending on their genotype, some patients may show excellent therapeutic response to capsaicin, whereas others (the “non-responders” [166]) may not. Furthermore, animal experiments suggest that TRPV1 expression may be decreased in some patients, for example those with diabetic polyneuropathy [169]. Therefore, testing the response of TRPV1 channels to capsaicin may help to select the patients who may benefit from the capsaicin therapy.

Last, there are conflicting preclinical results regarding the effect of TRPV1 blockade on cancer progression that still baffle researchers. For example, in preclinical experiments, the ablation of TRPV1-expressing afferents mediated by capsaicin or RTX blocked the growth of melanoma [57], had no effect on canine osteosarcoma [138], and accelerated the progression of 4T1 breast cancer metastasis to the lung (Figure 6, high-resolution version) [170,171,172]. Confusingly, in mice, the genetic inactivation of *Trpv1* prevented both bone colonization and lung metastasis formation by 4T1 breast cancer cells [173]. Adding to the confusion, the repeated activation of TRPV1-expressing sensory neurons was reported to promote tumor growth [174].

Whether these differences are cancer-pain-model-, species-, or cancer-related has far-reaching implications for drug development.


**Outstanding questions**


Is there a role for per os TRPV1 antagonists in cancer pain relief?Can site-specific (into the surgical wound) capsaicin or resiniferatoxin injections prevent the development of post-surgical pain in cancer patients?Can high-dose (8%) capsaicin patches relieve post-surgical (specifically, post-mastectomy) pain in cancer patients?Can topical capsaicin ameliorate localized pain in patients with oropharyngeal squamous cell carcinoma?What is the optimal dose of intrathecal resiniferatoxin at which it provides adequate pain relief with acceptable side-effects?Is epidural resiniferatoxin devoid of the side-effects of intrathecal administration?For TRPV1 knock-down mediated by siRNA given intrathecally, how does it compare to intrathecal resiniferatoxin?Does capsaicin/resiniferatoxin desensitization affect cancer growth?

## Figures and Tables

**Figure 1 cancers-16-00648-f001:**
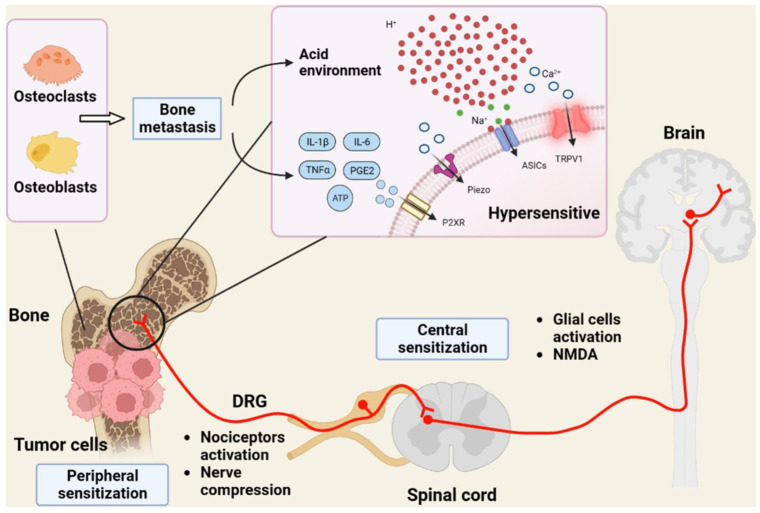
The complex etiology of bone cancer pain. Tumor cells that metastasize to the bone create an acidic environment that stimulates TRPV1 and other proton-sensitive channels in sensory afferents, like acid-sensitive ASICs. In turn (not shown), TRPV1-expressing nerve terminals release CGRP, a known facilitator of tumor cell growth. DRG, dorsal root ganglion; IL-6, interleukin-6; TNFα, tumor necrosis factor-α; PGE2, prostaglandin E2; P2XR, purinergic P2X receptor; Piezo, mechanosensitive ion channel. Figure reproduced with permission from [27].

**Figure 2 cancers-16-00648-f002:**
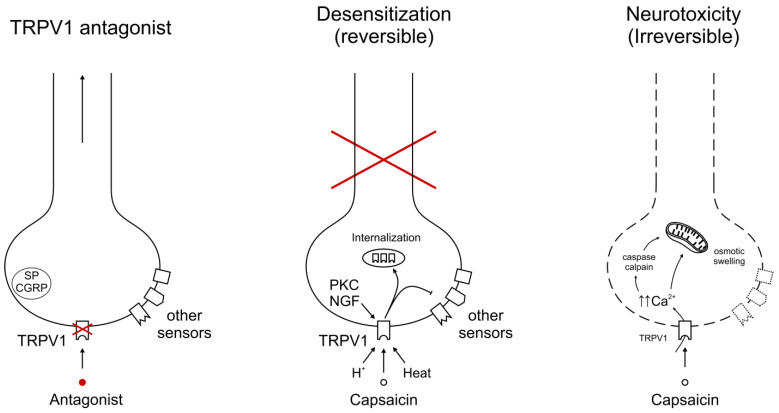
Molecular mechanisms of cancer pain relief, targeting TRPV1. Small-molecule TRPV1 antagonists block the channel protein only, leaving other pain-sensing targets functional. This approach will work only if cancer pain is mediated by an algesic compound that acts directly via TRPV1. By contrast, the chemical defunctionalization of TRPV1-expressing sensory afferents silences the whole neuron. This action can be either reversible (traditionally termed “desensitization”) or irreversible. A high-dose (8%) capsaicin patch is a representative example of desensitization. Intrathecal RTX (a “molecular scalpel”) may irreversibly ablate the central terminals of capsaicin-sensitive afferents for permanent pain relief. The cross in red indicates disruption by capsaicin of sensory information from the periphery (“capsaicin desensitization”).

**Figure 3 cancers-16-00648-f003:**
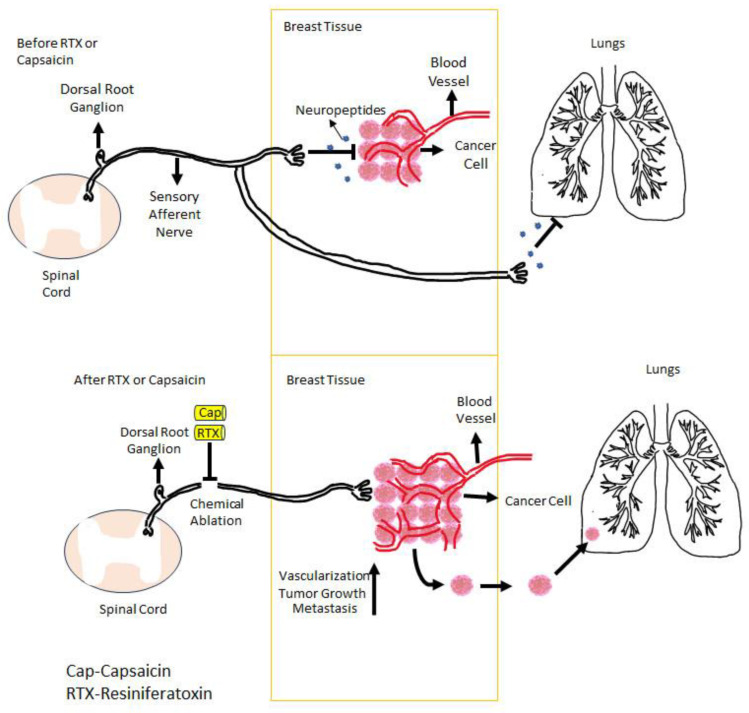
In breast cancer, TRPV1-expressing sensory afferents negatively control the vascular supply (neovascularization) of the tumor through the neuropeptides that they release. If these afferents are ablated either by capsaicin or RTX, this negative control is lost, and the tumor is starting to grow and metastasize. Figure courtesy of Mertay Şimşek and Nuray Erin (Akdeniz University, Turkey).

**Figure 4 cancers-16-00648-f004:**
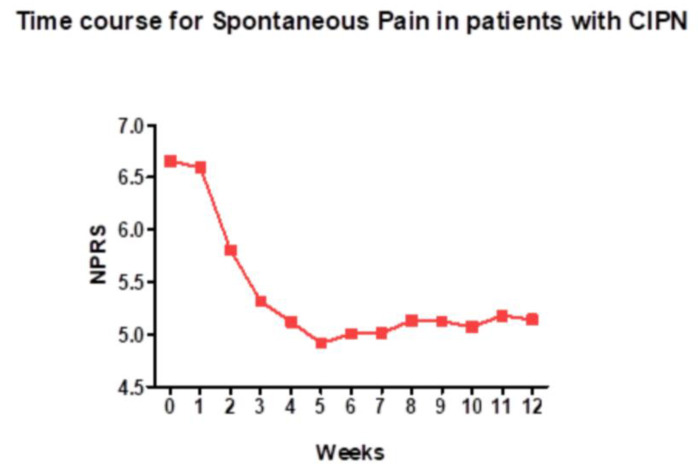
A high-dose (8%) capsaicin patch relieves spontaneous pain in patients with chemotherapy-induced peripheral neuropathy. NPRS (Numeric Pain Rating Scale) is a numeric version of the visual analog scale, in which patients select a number between 0 and 10 that best reflects the reduction in pain (10, no change; 0, complete loss of pain). Figure courtesy of Dr. Praveen Anand, Imperial College, London, UK.

**Figure 5 cancers-16-00648-f005:**
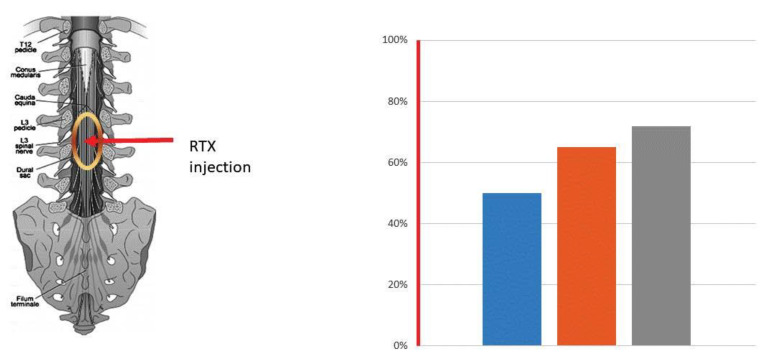
Cancer pain relief mediated by intrathecal RTX, 13 µg. The red arrow indicates the site of RTX injection between L3 and L4. The oval sign shows the spread of RTX in the spinal cord. RTX was injected under general anesthesia (1.5 h). An i.v. opioid was administered for the residual acute burning pain evoked by the RTX injection. The bars represent a reduction (in %) in the average NPR Score (50%, blue column), the 7 brief pain inventory (65%, brown column), and daily oxycodone use (72%, grey column) after RTX injection. Data are from a middle-age male patient with supraglottic squamous cell carcinoma metastasizing to the pelvic bone [145].

**Figure 6 cancers-16-00648-f006:**
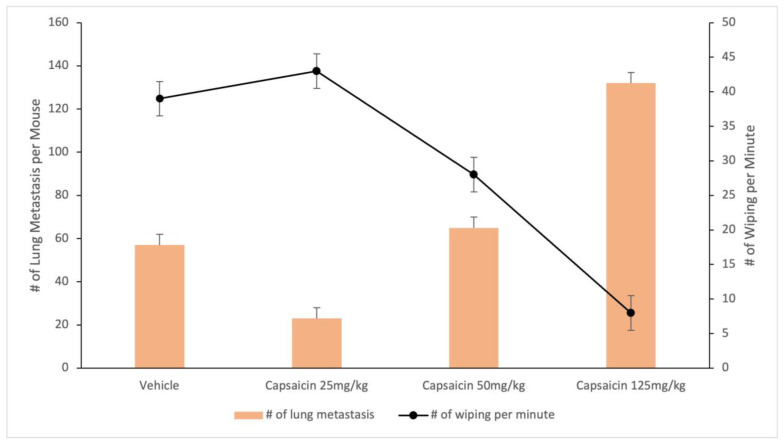
Desensitization of TRPV1-expressing afferents via s.c. capsaicin facilitates lung metastasis spread in a murine model of breast carcinoma. Data are from [124].

**Table 1 cancers-16-00648-t001:** Cancer pain; cancer pain treatments in adult patients—briefly.

**Cancer Pain**	
**Acute Pain in the Cancer Patient**	
the cause is the tumor itself (perforatio, exulceratio, ileus, acute organ compression, pathological fracture, increased intracranial pressure, hypercalcaemia, transverse spinal cord lesion, acute postoperative, etc.)	
2. pain during antitumor treatment (biopsies, blood sampling, surgery, examinations, acute, toxic neuropathy, burns, ulcers from radiation therapy, etc.)	
3. independent of the tumor (spasm caused by kidney stones, etc.)	
**Chronic Cancer Pain (Nociceptive, Neuropathic, Mixed)**	
the cause is the tumor itself	
2. it develops as a result of antitumor treatment (surgery, chemotherapy, radiation therapy)	
3. independent of the tumor (migrain, coxarthrosis, etc.)	
4. breakthrough pain	
**Cancer Pain Treatment in Adult Patients—Briefly**	
**Cancer Pain Treatment (Always Multimodal, Individual)**	
causal/palliative treatment (surgery, chemotherapy, radiation therapy +/− psychological support, +/− rehabilitation treatment	
2. WHO analgesic ladder + magamenet of (opioid) side effects	
3. +/− adjuvant therapy	
4. neurolytic nerve block	
5. intrathecal drug administration, spinal cord stimulation, cordotomy	
ad 2. WHO analgesic ladder	
**Pain Intensity (Numeric Rating Score)**	**Type of Analgesics**
1–3 (4)	minor analgesics (paracetamol, metamizol, NSAIDs)
4–6	weak opioids (tramadol, DHC, etc.)
>6	strong opioids (morphine, oxycodone, hydromorphone, fentanyl, buprenorphine, methadone, etc.)
ad.3. +/− adjuvant therapy	
**Type of Cancer Pain (Main Types)**	**Type of Adjuvant Therapy**
neuropathic, mixed (nociceptive + neuropathic)	anticonvulsive drugs (pregabalin, gabapentin), TCAs (amitriptylin, clomipramin), duloxetine, clonidine
2. bone metastasis, primary bone tumor	corticosteroid + antineuropathic drugs, calcitonin, bisphosphonate compounds
3. tumor compression symptoms, pressure increase within the parenchymal organ, lymphoedema, etc.	corticosteroid
4. skeletal muscle spasms	antispasticity agent (tizanidin, etc.)
5. colic	
6. excruciating urge to urinate, tenesmus	haloperidol

**Table 2 cancers-16-00648-t002:** Representative examples of chronic pain syndromes caused by solid tumors. GI, gastrointestinal.

Damage to the skeletal system	generalized bone pain due to metastatic disease and/or bone marrow infiltration pathologic fractures
Infiltration by solid tumors of visceral organs	shoulder pain based on diaphragmatic infiltration diffuse peritoneal disease (carcinomatosis) epigastric pain for upper GI tumors abdominal pain caused by liver metastasis obstruction of the ureter
Infiltration by solid tumors of soft tissue	infiltration of oropharyngeal tissues by head-and-neck squamous cell carcinoma
Nervous system involvement	headache due to leptomeningeal carcinomatosis intracerebral metastasis radiculopathy (e.g., vertebral or spinal lesions) plexopathy (e.g., branchial due to Pancoast tumor) peripheral nerve damage, including polyneuropathy caused by chemotherapy

**Table 3 cancers-16-00648-t003:** Types of chronic pain caused by hematological malignancies with representative examples.

nociceptive pain	superficial somatic deep somatic visceral	mucositis bone marrow expansion by neoplastic cells hepatosplenomegaly (stretches the capsule)
neuropathic	peripheral central	neuopathy due to paraproteins or amyloid CNS involvement
mixed		

**Table 4 cancers-16-00648-t004:** Clinical analgesic efficacy of epidural RTX (15 µg or 25 µg) assessed at a 30%, 50%, and 70% decrease in pain (average and worst pain) from the baseline NPRS score. Data are from [142].

% Decrease in Pain from Baseline	Average Pain	Worst Pain
30%	65% of study participants	47% of study participants
50%	35% of study participants	29% of study participants
70%	23% of study participants	18% of study participants

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
