# Peer review of "Targeting TRPV1 for Cancer Pain Relief: Can It Work?"

_cancers, 2024, doi:10.3390/cancers16030648_

Round 1

Reviewer 1 Report

Comments and Suggestions for Authors

The Author(s) have made a good efforts to gather the information and summarize the role analgesic potential of TRPV1 antagonism and sensory afferent desensitization in cancer patients. However, the author(s) need to work on the following comments to make the manuscript comprehensive.

1. The introduction of the manuscript needs to be revised and elaborated.

2. The author(s) needs to add more, latest and relevant references in the manuscript.

3. The references of the paper needs to be re-arranged and written according to the manuscript guidelines.

4. No self prepared (original figure) has been included by the authors, all figures are adopted. Please add more self prepared figures.

5. Revise the Figure 3, use high resolution figures and add more and original figures (self prepared).

6. Add self prepared tables, mention the role and participation TRPV1 in cancer pain.

7. Mention interaction between TRPV1 and tumor microenvironment.

Comments on the Quality of English Language

1. The manuscript need to improve the English and scientific language.

Reviewer 2 Report

Comments and Suggestions for Authors

Szallasi gives a review about whether targeting TRPV1 is effective in alleviating cancer pain.  At first, a schematic diagram in Fig. 1 demonstrates that bone cancer pain is complex in etiology.  Then, it is shown from published experimental data that capsaicin patch relieves spontaneous pain (Fig. 2) while TRPV1 desensitization results in lung metastasis spread potentiation (Fig. 3).  This review article seems to be interesting and timely in clinic.  There are several points that should be addressed and could help improve this manuscript, as follows:

Major points:

1.     Fig. 1: in the text, there is no description about “IL-1β, IL-6, TNFα, PGE2, ATP, P2XR and Piezo” that appear in this figure.  It may be better to briefly explain them in the legend of Fig. 1.  Does “NMDA” in this figure mean “NMDA receptor activation”?  If so, it would be better to state this fact.

2.     The text (Time course ...) written in large and bold letters just above Fig. 2 should be given as a title in the legend of Fig. 2.

3.     Section 4: is it possible that the capsaicin contained in the capsaicin patch acts as an analgesic by suppressing the action potential of the sensory nerves that transmit pain?  It is likely in the capsaicin patch that capsaicin exists at a high concentration in the vicinity of nerve fibers.  Capsaicin has an ability to not only activate TRPV1 but also inhibit voltage-gated sodium channels.  For information on the sodium channel inhibition, please refer to the paper (doi.org/10.1016/j.lfs.2013.01.011) and the cited papers listed in it.  Please give a comment on this fact, if the sodium channel inhibition is possible in the capsaicin patch.

4.     The sentences written just above Fig. 3 should be put in the legend of Fig. 3.  It should also be stated in the legend whether the reproduction of Fig. 3 has permission.  

5.     Is “TABLE” in line 313 given in a table?  Is “Outstanding questions” a table title?  If so, this should be given in a Table style and this content should be cited as a Table in the text.

Specific points:

1.     Line 96: not “stimulate” but “stimulates”.

2.     Lines 104 and 105: it is not necessary to repeatedly define “CINP” (see lines 40 and 41).

3.     Line 141: please expand “NPRS”.

4.     Lines 163 and 180: please explain shortly about “pregabalin” and “amitrypyiline”.  By the way, not “amitrypyiline” but “amitriptyline”?  Please check this point.

5.     Line 262, 274-283, 305 and 316-328: not “resiniferatoxin” but “RTX”.

Comments on the Quality of English Language

There is no problem about the Quality of English Language.

Reviewer 3 Report

Comments and Suggestions for Authors

I think the manuscript provides unbalanced information about associations between cancer pain and TRPV1.

Major comments

#1. The author needs to provide more comprehensive mechanism of cancer pain (inflammatory, neuropathic, and cancer-specific mechanisms) in part 1 or 2 (Page 1 to 3).

#2. Summary of “available treatment options” of cancer pain should be included in Part 2 or 3 (Page 7, Line 267). Then the author can focus treatment involved in TRPV1. There are only capsaicin and resiniferatoxin.

#3. I do not agree with the author that “cancer pain in rodents interests only scientists but cancer pain in companion dogs is an important issue in veterinary medicine” (Page 6, Line 211). Please provide reference.

#4. I do not understand why intrathecal resiniferatoxin must be done under general anesthesia (Page 6, Line 250). Please provide reasons.

#5. I do not understand why “epidural resiniferatoxin should be devoid of the adverse effects” (Page 7, Line 282). Please provide reasons.

#6. I do not agree with the author that “rodent models are good for acute pain, but not so good for chronic pain like cancer pain” (Page 7, Line 288). Please provide reasons and references.

#7. I do not see main conclusion.

Minor comments

#1. I do not think the p values provide meaningful information (Page 4, Line 142, 143).

#2. Please use paragraph more logically (Part 4 to 7).

#3. Figure 2 needs error bar.

Comments on the Quality of English Language

Paragraph is not used logically. Moderate editing of English language required

Round 2

Reviewer 1 Report

Comments and Suggestions for Authors

The author(s) although have tried to improve the manuscript and worked on few comments but is not yet enough. 

But the author(s) need to improve to the article to get it published.

1. Add novel literature.

2. Add novel and own figures.

3. Figure 3 still has no good resolution.

4. Author(s) should check quality of own previously published manuscripts on TRPV1.

Comments on the Quality of English Language

Minor editing of English language required.

Reviewer 2 Report

Comments and Suggestions for Authors

This revised manuscript has been largely amended according to my comments.  There are only minor points that should be taken into consideration before publication, as follows:

1.     There are no Keywords in line 22.  Please amend this point.

2.     Line 24: “Introduction” should be “INTRODUCTION”.  Please see the other subtitles.

3.     Line 42: TABLES 1 and 2 should be placed near this line, not on pages 11 and 12.

4.     Line 136: not “mechanosensitive ion receptor” but “mechanosensitive ion channel”?  Please check this point.

5.     Line 165: please define “DRG”.

6.     Line 301: TABLE 3 should be placed near this, not on page 13.

7.     Line 306: please put “, USA” following “CA”.

8.   The resolution of Fig. 3 should be increased.

Reviewer 3 Report

Comments and Suggestions for Authors

Comments and Suggestions for Authors

The manuscript is improved.

Minor comment

#1. Summary of “available treatment options” of cancer pain is still missing in Part 2 or 3.

Comments on the Quality of English Language

Moderate editing of English language required.
